# "It's All Bread, All the Way Down": *The Baby-Sitters Club Club* as Hyperfanfiction

Suzanne Scott

Department of Radio-Television-Film, The University of Texas at Austin, Austin, TX 78712, USA; suzanne.scott@utexas.edu

**Abstract:** In February 2016, co-hosts Jack Shepherd and Tanner Greenring launched their comedy podcast *The Baby-Sitters Club Club*. The "joke" at the center of the podcast was, of course, that two adult, cishet, white men were exhaustively recapping and dissecting a book series from the 1980s and 1990s that was predominantly popular with adolescent girls. What began as a podcast designed to poke fun at the co-hosts' serious "fannish" analysis of a nostalgic series of novels for girls, evolved into an elaborate string of "fan" theories, literary close readings, and (inter)textual expansion. Building on Paul Booth's discussion of hyperfans, this article theorizes the absurdist worldbuilding, mythology and character development, and intertextual play performed by the hosts of TBSCC as a form of hyperfanfiction.

**Keywords:** fanfiction; hyperfan; hyperfanfiction; gender; affirmational fandom; transformative fandom

## 1. Introduction

"Claudia's wearing a bra now, and the way she talks, you would think that boys had just been invented."—Jack Shepherd's first official sign-off for *The Baby-Sitters Club Club* podcast (7 March 2016).

"Baby nation, round off the corners of your bedroom, drown all of your dolls, call your Senator and demand your right to bear time, and do not forget even in these trying times to let Daddy love you as much as I do. Baby nation, remember the Delaneys, remember the Trip man, remember Boo-Boo, and take your dream horse through that maze. Claudia's wearing a bra now, and the way she talks, you would think that boys had just been invented."—Jack Shepherd's sign-off for the final podcast episode covering the main *Baby-sitters Club* book series (15 November 2018).

In February 2016, co-hosts Jack Shepherd and Tanner Greenring (both formerly of BuzzFeed) launched their comedy podcast *The Baby-Sitters Club Club*. The "joke" at the center of the podcast was, of course, that two adult, cishet, white men were exhaustively recapping and dissecting a book series from the 1980s and 1990s that was predominantly popular with adolescent girls. Reading and discussing one novel in Ann M. Martin's *The Baby-sitters Club* book series per week, *The Baby-Sitters Club Club* (hereafter *TBSCC*) ultimately ran over four years, comprehensively covering all 131 books in the original *Baby-sitters Club* series as well as all of its spin-off novel series[1] and ancillary products such as film and television adaptations and a 1996 interactive CD-ROM.

As the lengthy cumulative episode sign-off quoted above suggests, what began as a podcast designed to poke fun at the co-hosts' serious "fannish" analysis of a nostalgic series of novels for girls evolved into an elaborate string of absurdist "fan" theories, literary close readings, and (inter)textual expansion. The term "fan" is pointedly placed in quotation marks here not to imply that Shepherd and Greenring are not "real" fans. On the contrary, much of the podcast's content revels in the hosts' deep fannish investments in everything from Heidegger to the *Hellraiser* franchise, and it is emphasized throughout early episodes





that Shepherd encountered *The Baby-sitters Club* book series as an adolescent through a female cousin. Nor am I interested in litigating if Greenring and Shepherd's fan affect for *The Baby-sitters Club* book series is genuine or performative, or if or how that affective stance shifts over time. Like Bronies (adult male fans of *My Little Pony*) or podcaster predecessors *The Gilmore Guys* (two men watching and commenting on every episode of *The Gilmore Girls*) before them, the co-hosts of *TBSCC* unquestionably derive value and visibility from the gendered disconnect at the center of their podcast's premise, including coverage in high profile media outlets like *Vogue* (Ruiz 2020), but this doesn't mean they aren't legitimate "fans". Rather, I place "fan" in quotes here to gesture to the productive terminological and conceptual instability that a podcast like *TBSCC* provokes. Similar to arguments made elsewhere by Hills (2014) and Booth (2015) that particular modes of fannish production can productively complicate binary conceptions of fanworks as either "affirmational" or "transformative" in nature, the "fanfiction" created through *TBSCC* podcast is interesting for these liminal qualities, but also because it explicitly challenges the majority of dominant and enduring scholarly presumptions about the creation and consumption of fanfiction. Building on Paul Booth's discussion of hyperfans (Booth 2015, pp. 75–100), this article positions the absurdist worldbuilding, mythology and character development, and intertextual play performed by the hosts of *TBSCC* as a form of hyperfanfiction.

## 2. On Fan Podcasting and Forensic Fandom

Before delving into how *TBSCC* constitutes a form of fanfiction, much less hyperfanfiction, it is necessary to briefly address the particularities of the medium it is delivered through, namely a weekly fan recap or rewatch podcast. There are, of course, fanworks that are derived from sonic media objects (music fans, fans of podcasts, etc.), as well as aural forms of fanfiction such as audiofic or podfic, or "fan fiction that's performed verbally, recorded, edited, and then shared in audio format" (Riley 2020). Likewise, there is a long history of aural storytelling, from radio plays to popular podcasts like *Welcome to Night Vale* (Bottomley 2015) that inspire their own fan works such as fanfiction or fanart.

Because the components of *TBSCC* podcast that resemble fanfiction (e.g., the exploration of character backstories or alternate realities, worldbuilding and mythology development, and so on) are not derived from a pre-existing piece of *The Baby-sitters Club* fanfiction, nor is it the core component of the podcast, it is more generative to locate *TBSCC* in the growing genre of fan recap or rewatch podcasts. Recap culture is not a new phenomenon, as sites like *Television Without Pity* and *The A.V. Club* built their brands around fan-oriented content that was "partly television criticism, partly entertainment" (Falero 2016). However, as fan recap and rewatch (or, in this case, re-read) podcasts have proliferated, it is important to acknowledge that, more so than many other forms of fan production, the fan-producers of podcasts "become 'characters' themselves, with discernible tics and tastes as well as their own fans" (Diffrient 2010, p. 107). This is certainly the case with *TBSCC* podcast, as Shepherd and Greenring's heightened performance of their "odd couple" dynamic is baked into the podcast's description: "A big dumb idiot [Greenring] and his brilliant, charming friend [Shepherd, clearly writing the description] discuss the classic novels of Ann M. Martin in chronological order."

Indeed, while many (like myself) were initially drawn to *TBSCC* podcast through our own fannish nostalgia for *The Baby-sitters Club* book series, the members of "Baby Nation" (the collective self-identification for fans of *TBSCC* podcast) must ultimately also become fans of Shepherd and Greenring (their dynamic as friends, their propensity for high theory and masculinized geek culture references, and so on) in order to find full fannish pleasure in the podcast. Building on Lauren Savit's article on episodic TV podcasts as a form of fan labor, which closes with a call for further research on "the identity politics of the hosts and how that affects the dynamics of the relationship between fans of the source text, fans of the podcast, and the hosts" (Savit 2020), Megan Connor's experience of *TBSCC* podcast bears this tension out. She notes, "I felt, in the moment of listening, a deep territorial nostalgia

and unhappiness that two men had staked what Savit calls a 'proto-fannish authority' over a text so entwined with girls' feminine and feminist identity" (Connor 2022, pp. 85–86). Although my own initial kneejerk response to the podcast's "masculinist, albeit playful, perspective" (p. 85) echoed Connor's, I ultimately found that my fan nostalgia for the source material was augmented by Shepherd's belabored applications of theorists like Jacques Lacan and Roland Barthes, as well as the litany of other geek culture references (ranging from tabletop games to deep cut *Star Wars* characters). Still, this gendered tension at the core of the podcast's premise, and its potentially polarizing effects, is vital to any consideration of the "fanfiction" generated through the podcast, and will be addressed in more detail below.

Like most television recap or rewatch podcasts, TBSCC developed recurring segments over time to structure the weekly novel recaps and analysis. Some of these segments, like the "Burn of the Week" (in which Shepherd and Greenring pick their favorite insult from the novel under discussion) were rooted firmly in the text. However, because *The Baby-sitters Club* novels were designed for an adolescent readership, were often thin on meaningful plot and character development, and at times notoriously/delightfully redundant (see: the obligatory and mostly unchanging exposition passage/chapter reintroducing the central characters and their roles in the club in almost every book in the series), there were clear textual limits to extensive close analysis of the novels' content without any creative expansion or absurdist "theorizing". This was compounded by the fact that podcasts, by design, offer a more "long-form space" for fan interpretation, and accordingly demands of audiences "a temporal commitment that far exceeds Twitter or other text-based digital media" (Florini 2019). This investment is rewarded, in the case of hypermythologized or mystery-oriented source material, by podcasts offering fan listeners a "sense of perusing through stacks of (sometimes vague or even conflicting) tomes of legend and lore" (Lynch 2018, p. 156).

In perhaps a parodic nod to similar recap podcasts with more mythologically dense or character-dynamic rich source materials, Shepherd strived in early episodes to perform literary criticism and analysis, playing up his role as the pedant scholar of the pair to Greenring's feigned disinterest. To offer just two early examples, in episode 5 of the podcast ("Dawn and the Impossible Three"), Shepherd reads deep religious symbolism into the difficult baby-sitting charges referenced in the novel's title, spiraling from a discussion of the holy trinity to the connection between the number 33 in numerology and flat earth theory. Likewise, in episode 3 ("The Truth About Stacey"), Shepherd makes a comically tortured conceptual leap from a fleeting textual reference to "Pauline's Fine Candy" to the "Pre-Frontal Cortex", or "the part of the brain that is involved in making sure you follow social conventions", noting "it's so fucking obvious" that the book is a rumination on and critique of the surveillance state.

Whether as a byproduct of the novels' relatively thin plotting, or as a satirical response to composing a "recap" podcast with minimal fodder for recapping, what started as mock "serious" and "scholarly" analysis of a decidedly unserious and delightfully straightforward text quickly evolved into a complex web of conspiracies and worldbuilding exercises that veered closer to the realm of fan theories and fanfiction than recaps. Over time, *TBSCC* self-consciously developed into a recap podcast for an imagined *Game of Thrones* style source text, with factions of living dolls, dinosaurs, sentient orbs, and soldiers with magnificent weapons all vying for control of the sleepy fictional town of Stoneybrook, Connecticut. As scholars like Tosenberger (2008) and Åström (2010) have noted of the television series *Supernatural*, seemingly subversive or outlandish fan readings (and the resulting production of fanfiction, from incest to male pregnancy) can often emerge structurally from the text's own narrative design. Similarly, the design of *The Baby-sitters Club* book series structurally supported many of the outlandish "theories" developed on *TBSCC* podcast, pointedly through its lack of development. To offer a specific example, "Amber Theory", or the notion that the fictional town of Stoneybrook is encased in amber, never allowing the girls to age or leave, emerged because the central characters of *The Baby-sitters Club* novels spend the

first 10 books in 7th grade and the subsequent 10 years and hundreds of novels perpetually stuck in the 8th grade. More generally, though, the lack of plot and character development throughout a given novel (or even over the whole series) and the demands of a weekly, hour-long, ad-supported podcast, created conditions that necessitated textual speculation and expansion.

In order to accomplish this, *TBSCC* created a conceptual bridge between fanfiction and forensic fandom. Mittell (2009) describes forensic fandom as being rooted in textual mystery and complexity, resulting in a narrative structure that "structurally encourages viewers to parse the show more than simply consume it" (p. 128). *TBSCC* podcast lovingly lampoons this "hyper-attentive mode of spectatorship" (Mittell 2009, p. 128) and accompanying fan "detective mentality, seeking out clues, charting patterns, and assembling evidence into narrative hypotheses and theories" (pp. 128–29) through the creation of absurdist theories. We can view this in one of two ways. First, we might read this forensic fan approach as a way of reconciling the aforementioned gender disconnect between the hosts and source material by approaching a decidedly feminine texts through a more affirmational (Obsession_inc 2009) or conventionally masculinized mode of engagement. Alternately, we might interpret the highly speculative form "forensic fandom" produced on *TBSCC* as mocking these male-dominated modes of engagement and embracing a more transformative and playful relationship to the source material.

To offer perhaps the most farcical example, in episode 41 ("Poor Mallory!"), Shepherd is in the midst of highlighting the novel's connection to Dickensian literature when Greenring interjects a close reading of his own:

> Shepherd: It's Dickensian in its scope, it brings in a vast array of new characters, many of whom are parodies and caricatures of, like, grotesque richness and opulence and cruelty. Like these characters Valerie and Rachel and someone called Nan White. Who's called Nan White?

> Greenring: Naan White. Guess what? Naan is a kind of bread, so is white.

> Shepherd: [snickers] Ok. That's what you captured there?

> Greenring: White bread. Naan bread. Maybe there's something there? I don't know.

> Shepherd: We both looked at Nan White, I wrote "Bleak House", you wrote "TWO KINDS OF BREAD!"

> Greenring: THIS LADY IS TWO KINDS OF BREAD, WHAT DOES THAT MEAN?!

Over the course of the episode, "Bread Theory" is born. Roughly 10 min later, Nan (Naan) White is mentioned again, with Greenring noting that the theory is "worth putting a pin in, at least". Five minutes later, Shepherd makes an offhand comment while discussing minor character Hannie Papadakis that "Papadum is a kind of bread", prompting a "Whoa!" from Greenring who notes that "This is shaping up into a thing. It's fresh, it's new, the yeast is still rising on this fan theory". Two minutes pass, and as discussion moves on to a minor character named Michael Hoffmeister, Shepherd deadpans "Do you want to google and see if Hoffmeister is some kind of a bread? Please don't", he immediately retracts as Greenring proclaims he's "deep into bread theory now" and promptly locates a bakery in Karlsruhe, Germany named Hofmeister-Brot.

Bread Theory would become a fan favorite recurring segment, generating both fan-made and official merchandise in the Wonder bread font and color story. By episode 48 ("Mary Anne Misses Logan"), both hosts were jokingly proclaiming that "It's all bread", or "Bread all the way down". On the same day that episode 41 of the podcast was released, an image was posted to the Baby Nation facebook group, the community hub for fans of the podcast, to commemorate the emergent theory: an instantly recognizable play on Agent Fox Mulder's iconic "I Want To Believe" poster from *The X-Files*, with the alien spaceship replaced with a loaf of bread (see: Figure 1). If absurdism is, at its core, "a conspicuous

discrepancy between pretension or aspiration and reality", and by extension, "the perpetual possibility of regarding everything about which we are serious as arbitrary" (Nagel 1971, p. 718), this poster and Bread Theory more broadly can be viewed as emblematic of the podcast's absurdist approach to fan engagement. However, I would suggest it is precisely because so much of the podcast's humor is derived from parodying both affirmational (forensic fandom) and transformative (fanfiction) modes of fan engagement that it is worth exploring both the pleasures and potential disciplinary functions of this output.

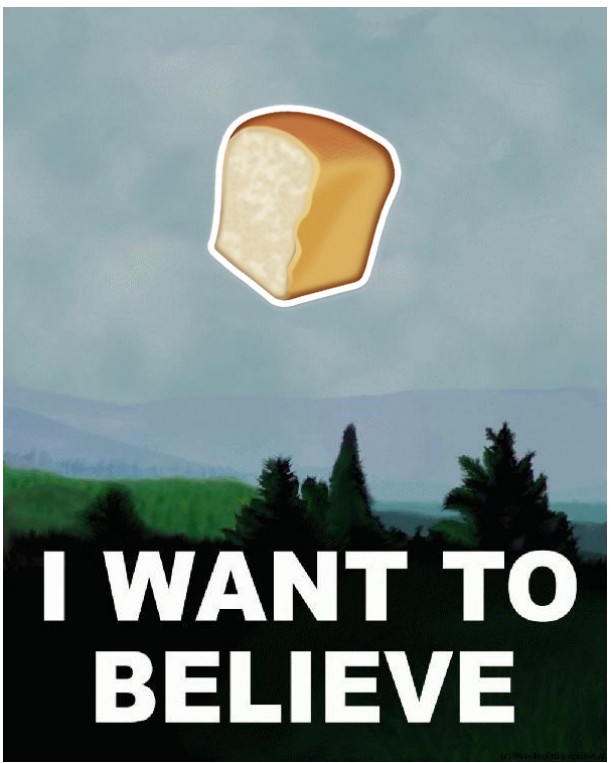

**Figure 1.** Fanart posted to the "Baby Nation" facebook fan group, the same day that "Bread Theory" was introduced on the podcast.

### 3. *TBSCC* and/as Hyperfanfiction

In the 2016 trailer for *TBSCC* podcast, Shepherd invited potential listeners to "read along with us as we pick the strands out of this delicate tapestry that Ann M. Martin has woven for us, in order to weave a brand-new pattern of our very own". This description immediately brings to mind descriptions of fanfiction from within fan studies, and in particular Stanfill's (2015) framing of fan remixing practices and transformative works as "spinning yarn with borrowed cotton" (p. 131). With this said, the "fanfiction" produced within the *TBSCC* podcast contradicts many of the prevailing understandings and theorization of fanfiction as a practice. As Stanfill (2015) notes, understanding fan works like fanfiction as "spinning yarn" is "an aptly gendered metaphor for deeply feminized forms of labor" (p. 132). It is precisely because fanfiction has long been approached as a practice dominated by women writers and readers, and has further been theorized as "a literature of the subordinate" (De Kosnik 2006, p. 72) that there are immediate difficulties in framing what the white, cishet hosts of *TBSCC* are doing as fanfiction. Importantly, there are also longstanding issues with fan scholars presuming both the whiteness and straightness of fans generally (see: Wanzo 2015; Warner 2015; Pande 2018; Stanfill 2018; Woo 2018; Pande 2020), and fanfiction writers and readers specifically, frequently treating these identity markers as an unacknowledged "default" identity. The primary point I wish to make is that fanfiction has historically been understood as a practice dominated by those who self-identify as women, without erasing broader issues about the similar fannish privileges

that white straight women retain in these scholarly and cultural accounts of the gendered nature of fan engagement.

This difficulty in labelling the podcast's creative play as "fanfiction" is compounded by the hosts' forensic fandom approach to generating new stories from the source text, which overwhelmingly focuses more on crafting a denser mythology and worldbuilding than the character/character dynamic exploration that has historically driven and dominated fanfiction production. Accordingly, the analysis of the fannish narratives generated within *TBSCC* podcast offered below suggests a need for different theoretical approaches to fanworks that have a more tenuous relationship to the sorts of fan identities, communities, and practices that have historically been studied.

Drawing on Jean Baudrillard's theory of hyperreality, Booth (2015) introduced the term "hyperfan" to address the "twinned representational parodies of fandom [that] have appeared in film and television in order to both generate more fandom and to attract non-fan audiences" (p. 75). Importantly, Booth argues that:

> "these twinned representations implicitly contrast dominant readings of 'bad' fandom (excessive, transformative, feminine) with dominant readings of 'good' fandom (appreciative, supportive, commercial) through a contrast between the two that attempts to discipline fandom into one particular identity". (pp. 75–76)

Because the content of *TBSCC* reflects both an awareness of these twinned representations and a persistent desire to mock both of these fannish poles, I would contend it is most productive to approach the textual expansion produced by the hosts as a form of hyperfanfiction. Whereas Booth's term is more focused on media representations of fans, and the disciplinary function they might serve, I would like to suggest through this analysis that these logics might be extended to performances of fannish textual expansion that exist outside fan communities of practice and have a more ambivalent relationship to a fan object. Although hyperfanfiction might serve a similar disciplinary function, it is important from the outset to distinguish it from trends that are more explicitly designed to shame predominantly feminized fan cultural practices such as writing and consuming fanfic and other transformative works. This notably includes instances when celebrities read homoerotic fanfiction featuring their character, or display slash fan art, on talk shows for the sole purpose of mockery (Jones 2014). The comedy that emerges from the hyperfanfiction produced by *TBSCC* in undoubtedly complex, and potentially disciplinary of both masculinized and feminized forms of fan engagement, but it is not meanspirited by design.

The analysis of three distinct examples of hyperfanfiction from *TBSCC* podcast below is designed to consider how understandings of fanfiction might be complicated if we examine more liminal (or even incidental or unintended) cases of fannish textual production. In many ways, the forms of fanfiction produced within *TBSCC* podcast are exemplary of longstanding theorizations of fanfiction that stress performance over text (Coppa 2006, p. 225), or approach fanfiction as "fragmentary ephemera" (Busse 2017, p. 148), albeit with an insular fan community of practice limited to the two hosts.

### 3.1. Case Study: Trackin' Jackie

"Trackin' Jackie" is typical of many of the recurring segments on *TBSCC* podcast that might be labelled as iterative forms of hyperfanfiction. Introduced in episode 21 ("Kristy and the Walking Disaster"), the podcast hosts begin as many fanfiction authors might: by using passing references in the text to riff on a minor character's potential backstory. In this case, the character in question is accident-prone baby-sitting charge Jackie Rodowsky, whom Shepherd and Greenring eventually decide "phases in and out of time". This passing remark would lead to an ongoing segment for almost all future appearances of Jackie Rodowsky in the novels, in which any time Jackie trips or knocks something over, he blips forward or backward in time, going on adventures and living whole lives before being ripped back to present day Stoneybrook and returned to his childhood body.

"Trackin' Jackie" was officially cemented as a recurring segment in episode 23 ("Jessi Ramsey, Pet-Sitter"), complete with an introductory audio sting of plucky sci-fi theremin

music and dialogue from the ABC television series *Lost* of protagonist Jack yelling "We have to go back, Kate! We have to go back!" This iconic bit of dialogue, drawn from the final moments of the notoriously complex and mystery-driven show's third season finale ("Through the Looking Glass"), is notable for the reveal that *Lost*'s narrative had shifted from a flashback to a flashforward structure. In addition to directly evoking the same time-bending logics that "Trackin' Jackie" was predicated on, aurally referencing one of the preeminent forensic fan objects of the past several decades also works to temper a feminized practice (fanfiction writing) with references to a masculinized text and mode of engagement.

This logic also extends to the fanfiction produced within the segment, which skewed towards stereotypically masculinized genres such as political thriller, action-adventure, and science fiction. For example, in the first full "Trackin' Jackie" segment, Shepherd and Greenring draw on context clues from the novel to inspire an alternate history in which Jackie lives an entire life, prevents the assassination to John F. Kennedy, and then blips back to his life in 1980s Stoneybrook only to realize that he is not in the utopic America he helped create. In future segments, Jackie would be reimagined as a stand-up comedian in gritty 1970s New York City (Episode 88: "Claudia Kishi: Live from WTSO!"), an explorer with the 1910 Terra Nova Expedition (Episode 29: "Welcome Back, Stacey!"), a colonizer of the moon (Episode 25: "Kristy and the Mother's Day Surprise") or the first astronaut to reach Mars in the year 2192 (Episode #58: "Keep Out Claudia"), and so on. In episode 38 ("Jessi's Baby-sitter"), Jackie even blips forward through time within the fanonical universe created by *TBSCC* podcasts hosts, encountering an apocalyptic version of Stoneybrook rich with references and inside jokes to the mythology created over the course of the podcast.

There are several ways in which we might approach "Trackin' Jackie" as an exercise in hyperfanfiction, or by offering a twinned representation of fan practices. Just as hyperfan representations can "obscure a more insidious representation of fandom, the one that more quietly and subtly depicts more 'proper' instantiations of fan activities" (Booth 2015, p. 76), "Trackin' Jackie" as a form of hyperfanfiction simultaneously revels in the desire to produce fanfiction and sidesteps its negative cultural associations through a masculinization of the practice. Though much fanfiction production centers on exploring underdeveloped minor characters, the centering of a minor male character as the epic protagonist in a book series dominated by female characters and skinning these exercises in more "appropriate" (read: culturally valued) genres safely creates a buffer between the segment and fanfiction writing as a practice. In this sense, we might position "Trackin' Jackie" and similar hyperfanfiction segments on *TBSCC* podcast as "both preserving and mocking" fanfiction writing as a practice, "in order to provide a more normal (i.e., disciplined) fan response" (Booth 2015, pp. 85–86). The mere fact that this functions as a recurring podcast segment serves to "discipline" the cultural production contained within it (e.g., this is typically a short, standalone[2] segment that is delivered late in an episode and mostly detached from the broader worldbuilding and mythologizing undertaken by the hosts). The segment is also disciplined (or rendered more "normative" and removed from the always already feminized associations of hyperfandom) by the lived identities of the hosts, and their conspicuous conceptual distancing from both the fan object ostensibly at the center of their podcast as well as fanfiction as a practice.

### 3.2. Case Study: A Time to Kilbourne

The most explicit (or, at the least, most conventional) example of fanfiction produced within *TBSCC* podcast emerged from a passing joke surrounding Shannon Kilbourne, a minor character from *The Baby-sitters Club* novels who served as an associate member and occasional baby-sitter. In episode 63 ("Jessi and the Awful Secret"), Shepherd notes that Shannon Kilbourne is introduced in book 11 ("Kristy and the Snobs") and then "just disappears until now". Greening corrects that she is mentioned in prior books, but "spoken about as a figure of legend", leading the hosts of *TBSCC* podcast to speculate in that "she is involved in some super-secret mission for the United States government" in order to

explain her infrequent appearances in the books. Shepard then admits: "Fun fact: I even wrote a short story called *A Time to Kilbourne* about that very possibility", leading to a long back and forth about how details from the novel under discussion points to the fact that Shannon is a "spook for the George H.W. Bush administration". Despite meeting four of the five definitional properties of fanfic outlined by Coppa (2017, pp. 2–14), these being 1. produced outside of the literary marketplace, 2. rewriting and transforming another story that is 3. owned by others, and 4. focused on a character rather than the storyworld, Shepherd's initial designation of *A Time to Kilbourne* as a "short story" rather than "fanfiction" is telling. The framing was not due to a lack of familiarity with practice, as the hosts mentioned the possibility of composing character-driven fanfiction in the very first episode ("Kristy's Great Idea"). This description is even more confounding when the "short story" is revealed to be only two paragraphs long, far more closely related to a "drabble" in fanfiction parlance.

A month after the first mention of *A Time to Kilbourne*, in episode 66 ("Dawn's Family Feud"), the topic of Shannon Kilbourne fanfiction comes up explicitly. After performing a close analysis of the character's appearance in the novel and how it connects back to her potential identity as a spy who's come in from the cold, Shepherd muses, "Do you think there's a Shannon Kilbourne standalone?", prompting Greenring to google and discover the character's tag on fanfiction archive AO3. As Shepherd loudly argues that he's composed the only such story, Greenring makes a noise of disgust, prompting Shepherd to narrate, "Tanner's making a face like he just saw something that really upset and hurt him". The offending content, it is revealed, was a description for a "charged" fanfiction story featuring Shannon, prompting snickers from both before Shepherd launches directly back into fannish analysis of the text and the subject is dropped.

This moment is exemplary of the sort of tension that is baked into the "hyperfan", and characterizes the podcast's distinct production of "hyperfanfiction" by extension, marked by two layers of presentation of fannish textual production: "one that the audience can mock and other that teaches the audience an appropriate manner of fannish behavior" (Booth 2015, pp. 93–94). The messaging that A03 fanfiction production is innately disturbing, while "short stories" that, although they might be rooted in the long history of fanfiction crossovers, actively distance the work of fanfiction from "feminized" practices through its generic framing as a spy thriller (or the evocation of Grisham 1989 courtroom drama via the title) remain "appropriate". Even with this point of differentiation, it is telling that it was only after *TBSCC* podcast had exhausted every single book in *The Baby-sitters Club* series and its spin-off series that *A Time to Kilbourne* was released as a special bonus episode on 7 June 2021. The hosts explicitly acknowledge this early in the episode, with Greenring noting that "Time has forced our hands and we are sort of being compelled to do this", gesturing to both the lack of any more canonical *Baby-sitters Club* content and the demands of the podcast's advertising partners to get an episode out. Shepherd plays off these pressures, suggesting that the episode represents the pair "going above and beyond" to "tie up loose ends". This dual framing concurrently positions the act of composing fanfiction as an embarrassing afterthought or necessary evil, and as a way for fans to transcend expectations of commitment to a text.

The brief reading that follows in many ways embraces the elements of podfic (a dramatic reading of a fanfiction story, complete with a generic spy thriller score and sound effects when appropriate), but also continues to play with conceptions of what "appropriate" fannish textual production might look like. *A Time to Kilbourne*, like all the hyperfanfiction produced within the podcast, mines comedy from the intersection of forensic fan approaches (exhaustive descriptions of Kilbourne's spy tech and weaponry) and the feminized banality of the source text (getting a call mid-mission for a baby-sitting job). So, although there are ample intertextual references to *The Baby-sitters Club* pepped throughout the fanfiction (e.g., the Bond-style villain of the piece is "The Phantom" of "Claudia and the Phantom Phone Calls", the repositioning of Claudia's genius older sister

Janine as an "M"-like figure, etc.), the fanfiction produced is not framed as an exercise in transformative textual production, but rather deeply affirmational textual exploration.

"This is the polar bears episode", Shepherd jokes, referencing a notorious mystery from the pilot of *Lost*. In this way, Shephard positions himself as both frustrated fan (wanting answers and ultimately composing his own via fanfiction) and all-knowing author. This emphasis on authorial intent, or the premise that *A Time to Kilbourne* is ultimately more an exercise in close textual analysis than fannish textual production, is reiterated throughout the episode. The episode description opened: "Five years ago, two men made a solemn promise: They would tell the story of Agent Associate Baby-Sitter Shannon Kilbourne the way Ann M. Martin always intended it to be told". Positioning themselves as "priests" or "avatars" who are merely relaying Martin's "wisdom" and story elements that are "deeply engrained in the text", the hosts of *TBSCC* develop a form of hyperfanfiction that exists between and pokes fun at both fannish textual reverence and resistance.

*3.3. Case Study: Ann M. Martin and Her League of Extraordinary Ghostwriters*

At its core, Booth presents "hyperfan" representations as an industrial strategy, provided to "discipline contemporary fandom into useable fan audiences" (Booth 2015, pp. 82–83). Accordingly, a major point of differentiation between industrial understandings and depictions of "normative" and "excessive" fandom hinges on a given fan's or fan practice's relationship to authorship and the authority it carries. Obsession_inc (2009) stresses authorial respect in their delineation between affirmational (industrially sanctioned) and transformative (unsanctioned) fan engagement, and Booth (2015) reiterates in his discussion of hyperfandom that more positive representations of "normative" fan identity tend to be marked by a "celebratory notion of the creator" (p. 96). In their production of hyperfanfiction focused on the primary author, ghostwriters, and publishing executives, the hosts of TBSCC podcast take a taxonomic approach in their depiction of fan/author relations, alternately mocking and celebrating the parasocial relationships that fans develop with content creators and executives.

Over time, the hosts' reverence for and mythology building around original series author Ann M. Martin, developed into a lengthy series of epithets that would open most episodes. For example, from episode 84 ("Kristy and Mr. Mom"), Shepherd opens the podcast by introducing their focus on the "classic novels" from: "Princeton's own Princess Annabelle Matthews Martin, sanctified, stormborn, Mother of Clocks and bane to bats, first of her name, last of her kind, last hope for humankind, and author of the great Sitter's Cycle". Because it would be too exhaustive to contextualize the origins of each of these distinct designations across dozens of novels and podcast episodes, it will suffice to say that these inside jokes all position Martin as a sort of deity and build mythology around her all-encompassing powers as an author. Likewise, the designation of "The Great Sitters Cycle" (perhaps meant to playfully evoke Wagner's Ring Cycle operas) elevates and canonizes the novels as high art.

Positioned on the opposite end of the spectrum to Martin was "The Leviathan". In a parody of the ways in which fans might direct blame for various creative decisions at "The Powers That Be", or industry executives, so that they can remain reverent of the creators behind their fan object, editorial director of the publisher for *The Baby-sitters Club* book series David Levithan was reimagined as a Lovecraftian horror "lurking in the shadowy basement of Scholastic" (episode 87, "Dawn and the School Spirit War"). Notably, David Levithan's first contact with *The Baby-sitters Club* novels came as "a 19-year-old intern working on the series, tasked with keeping a "bible" so that nothing would be mixed up or forgotten" (Doll 2012). This is noteworthy both because the presence of a "Bible" is typically reserved for fictional worlds with highly complex mythologies, and because this actively aligns "The Leviathan" with more canonical, affirmational, and forensic fan identities and practices. Within "The Leviathan" mythology, the "dusty tomes" of *The Baby-sitters Club* Bible are discussed by the podcast's hosts as alternately an object of desire and fear, as "seekers after this Bible know that everybody who has ever looked at it has

gone raving mad". The tension produced by the desire and danger to "peer into the maw of the Leviathan", narrativizes the "troubling binary" at the core of Booth's (2015) hyperfan theory, crafting a form of hyperfanfiction that mocks both fannish investment and immersion as well as more "disciplined" or knowledge-driven approaches (p. 77).

The hyperfanfiction surrounding Martin, "The Leviathan", and the most consistent ghostwriters for the original series and spin-off novels became such a recurring feature that it spawned official merchandise (see Figure 2) in the popular "&" t-shirt design aesthetic. Importantly, this design was originally conceived in 2001 by Amsterdam-based design studio Experimental Jetset as a piece of fan merchandise for the Beatles (e.g., "John&Paul&George&Ringo"), and has since become a staple of fan merchandise for a wide array of franchises and characters. This visualization of the podcast's author-centric exercises in hyperfanfiction understandably places Ann M. Martin at the top of the list, with The Leviathan occupying an appropriately shadowy presence in the "basement" of the line-up. Once *TBSCC* podcast moved into *The Baby-sitters Club* novels penned by Martin's "League of Extraordinary Ghostwriters" (discursively positioned as a sort of superhero team), a more taxonomic approach to depictions of fan/author relations emerged. Some were given epithets and backstories, others were only mentioned in passing. Although some, like "The Entity" (to describe the only co-authoring duo of the series), carried on the tradition of crafting vaguely fantastical or science fictional stories out of minimal information, the hosts most consistent and obsessive expressions of fan affect were reserved for frequent ghostwriter Peter Lerangis, also known as "Sweet Pete". These expressions of fandom can be read as both genuine (in the hosts' palpable delight when they arrive at a "Pete book") and performative (in the hosts' awkward sexualization of "Sweet Pete and his sweet sweet feet" and their repeated efforts to make contact over social media). It would be too easy to suggest that the hosts of TBSCC valorize and elevate "Sweet Pete" above other ghostwriters purely because he, too, is a white man deigning to devote time to a book series designed for young girls. Rather, because the hosts' discussions of Sweet Pete simultaneously veer the closest to hyperfandom of any of these instances of author-based hyperfanfiction and presents an outsized appreciation for more "normative" (e.g., masculinized) entries in the series, it is emblematic of how these "twinned" representations function.

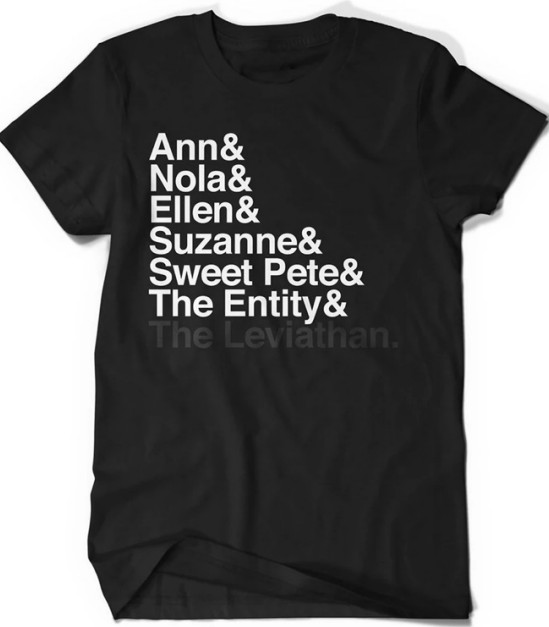

**Figure 2.** Official Merchandise from *TBSCC* podcast featuring the hosts' ongoing fascination with author Ann M. Martin and the various ghostwriters of the series.

## 4. Conclusions: Strange Bedfellows

It is precisely because all of the aforementioned forays into "fanfiction" production "presents obvious parodies of contemporary fan activities" (Booth 2015, p. 76) that we can classify it as "hyperfanfiction". However, rather than twinned representations that contrast a feminized/transformative "hyperfan" and a more "normative"/masculinized depiction of affirmational or forensic fandom, *TBSCC* podcast frequently collapses these categories in their iterative and fragmentary fannish textual production. Although a case could certainly be made for this collapse serving a similar disciplinary function as the one described by Booth (2015), we can alternately read these instances of hyperfanfiction as lovingly poking fun at both the forensic and creative impulses of fan audiences equally. Regardless, it is difficult to disregard the gendered disconnect that sits at the heart of the podcast's appeal, as well as its performances of fan engagement. Because hyperfan representations ultimately strive to "discipline fan audiences into behaving like proper fans—and, in all of these cases, the proper fan is masculine and knowledgeable, not feminine and emotional" (Booth 2015, p. 94), it is impossible not to engage the similar biases that underpin hyperfanfiction as a practice.

After running through all content even peripherally related to *The Baby-sitters Club*, and dabbling in other nostalgic book series aimed at young women (like *Sweet Valley High*), Shepherd and Greenring ultimately pivoted to new content. The resulting podcast, *Strange Bedfellows*, which initially focused on romance novels but has since expanded to cover a variety of romance media, is similarly premised on the demographic disconnect between the fan object and the hosts. In the premiere episode of Strange Bedfellows, Greenring jokingly discusses the impetus behind this rebranding, noting that the pair decided "what we need to do is we need to invade a space that's not for us and make it our own in true straight, white man fashion". "Fully manspread", retorts Shepherd. Because so much of the humor produced within Shepherd and Greenring's podcasts is rooted in parody, as is Booth's theorization of hyperfandom, it is easy to write of these moments of comedy as self-aware, or even self-reflexive, rather than ultimately modelling an appropriate fan identity.

Some scholars have positioned podcasts as being particularly well equipped to "create enclaved networked spaces" where black and minority fans "can engage in fandom, free from the discomforts and hostilities that come from operating in normatively white fan spaces" (Florini 2019), and this is certainly the case. When approaching a "fan" podcast like *TBSCC*, however, it is necessary to grapple with its internal ambivalences, tensions, and contradictions. Just as Tiffe and Hoffmann (2017) have suggested that podcasts open up a rich space to interrogate beyond bodily presence to consider who is allowed to sonically "take up space" in culture (p. 116), *TBSCC* poses similar questions about who is allowed to "take up space" in fan discourses and practices. Although the ongoing exploration of fanfiction as a practice that centers minority voices and creators is vital, it is equally essential to consider when and how fanfiction as a mode of cultural production either replicates biases or inequities or actively alienates the very demographics it claims to champion. More liminal cases, such as the fannish cultural production forwarded by *TBSCC* podcast, challenge longstanding presumptions about the production and reception of fanfiction, and accordingly offer a uniquely rich space to consider fanfiction as a cultural practice and how it might shift when placed into dialogue with more historically forensic modes of fannish analysis.

**Funding:** This research received no external funding.

**Institutional Review Board Statement:** Not applicable.

**Informed Consent Statement:** Not applicable.

**Data Availability Statement:** Not applicable.

**Conflicts of Interest:** The author declares no conflict of interest.

## Notes

1    Additional books in *The Baby-sitters Club* series covered by *TBSCC* podcast include canonical expansions on the main series, such as the *Super Specials* (15 books), *Mystery* (36 books) and *Super Mystery* (4 books) series, *Friends Forever* (12 books), *Portrait Collection* (6 books), and *Readers' Request* (3 books), as well as spin off series *The California Diaries* (15 books) and *Baby-sitter's Little Sister* (122 books).

2    Although "standalone" can be used to describe a fanfiction story, the term is more likely to be attached to comic books, television episodes, or computer software than fannish media production.

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
