# Peer review of "“It’s All Bread, All the Way Down”: The Baby-Sitters Club Club as Hyperfanfiction"

_humanities, doi:10.3390/h11050103_

Round 1
Reviewer 1 Report
Review of “It’s all bread, all the way down”
Humanities
Thank you for the opportunity to review “It’s all bread, all the way down”: The Baby-Sitters Club Club as Hyperfanfiction.” I enjoyed reading it – it is very well written and argued. It makes an important and significant point about the way fanfiction tropes are woven into this podcast, both reinforcing and (perhaps) subverting the tendencies of affirmational and transformational fan practices. I found this a compelling argument.
I particularly applaud the author for staying focused on the podcast and the immediate theories that help undergird their analysis – there are not long stretches about unrelated elements. The article is taut, clear, and precise. The methodology and examples are sound.
If I could add one (minor) element, it would be to insert within the article a short paragraph or two that used other podcasts as contrasts (perhaps revealing the same issue, or perhaps not). I’m surprised Welcome to Nightvale wasn’t mentioned, just because it is foundational in studies of podcasts and fandom (although I do recognize it is not doing the same thing as TBSCC here). One podcast that might be useful to bring into consideration is Aack Cast by Jamie Loftus – a female-led podcast that focuses on Cathy and brings some fanfiction elements into the podcast.
Overall, I truly appreciated the clear and specific method, literature, and analysis in this piece.
Author Response
First and foremost, a hearty thank you, both for the quick turnaround and for the generous and helpful comments on the article. In the revised draft, I have included references to/scholarly literature on "Welcome to Night Vale." As the reviewer notes, it's helpful to point to this broader body of literature that's been a cornerstone of podcast studies, but it's not a precise analog for the type of "storytelling" I'm discussing here. Because of the very limited turnaround time for the revisions, I was unable to consume and include the Aack Cast, though I am absolutely planning to check it out. I am hopeful that others, like the reviewer, will be able to make connections to how this trend manifests in other podcasts and expand on this in future articles.
Reviewer 2 Report
This is a really interesting case study and a nuanced, well considered unpacking of how the gender distinctions in fan creation behavior (the recap/video essay/podcast forensic fandom culture vs. the fan fiction creation transformative works culture) can potentially be collapsed and complicated when these cultures collide. The article applies the lens of Booth’s “hyperfan” to the analysis in this new context skillfully, and integrates a solid body of research to the analysis. The article makes a valuable contribution to the field of fan studies and fan fiction studies in its consideration of the shifting boundaries of gendered fan performance and fan creation as fan culture and content creation becomes increasingly mainstreamed and ventures into new digital forms.
Some minor copy editing issues for revision:
Line 9 – capitalization error in “what began”
Line 247 – missing “are” in “that more explicitly designed”?
Line 248 – punctuation error – should be a comma rather than a period after predominantly? Or perhaps the sentence revised to avoid excessive run-on.
Author Response
First and foremost, a hearty thank you, both for the quick turnaround and for the generous and helpful comments on the article. I have made all of the corrections, thank you for the keen eye/copyediting notes!